# Adipocyte *Utx* Deficiency Promotes High-Fat Diet-Induced Metabolic Dysfunction in Mice

**DOI:** 10.3390/cells11020181

**Published:** 2022-01-06

**Authors:** Fenfen Li, Shirong Wang, Xin Cui, Jia Jing, Liqing Yu, Bingzhong Xue, Hang Shi

**Affiliations:** 1Department of Biology, Georgia State University, Atlanta, GA 30303, USA; fli3@gsu.edu (F.L.); swang38@student.gsu.edu (S.W.); xcui@gsu.edu (X.C.); jjing@gsu.edu (J.J.); 2Division of Endocrinology, Diabetes and Nutrition, Department of Medicine, University of Maryland School of Medicine, Baltimore, MD 21201, USA; LYu@som.umaryland.edu

**Keywords:** epigenetics, obesity, adipose tissue, *Utx*

## Abstract

While the main function of white adipose tissue (WAT) is to store surplus of energy as triacylglycerol, that of brown adipose tissue (BAT) is to burn energy as heat. Epigenetic mechanisms participate prominently in both WAT and BAT energy metabolism. We previously reported that the histone demethylase ubiquitously transcribed tetratricopeptide (*Utx*) is a positive regulator of brown adipocyte thermogenesis. Here, we aimed to investigate whether *Utx* also regulates WAT metabolism in vivo. We generated a mouse model with *Utx* deficiency in adipocytes (AUTXKO). AUTXKO animals fed a chow diet had higher body weight, more fat mass and impaired glucose tolerance. AUTXKO mice also exhibited cold intolerance with an impaired brown fat thermogenic program. When challenged with high-fat diet (HFD), AUTXKO mice displayed adipose dysfunction featured by suppressed lipogenic pathways, exacerbated inflammation and fibrosis with less fat storage in adipose tissues and more lipid storage in the liver; as a result, AUTXKO mice showed a disturbance in whole body glucose homeostasis and hepatic steatosis. Our data demonstrate that *Utx* deficiency in adipocytes limits adipose tissue expansion under HFD challenge and induces metabolic dysfunction via adipose tissue remodeling. We conclude that adipocyte *Utx* is a key regulator of systemic metabolic homeostasis.

## 1. Introduction

Obesity is associated with various metabolic disorders, including insulin resistance/type 2 diabetes and fatty liver disease [1]. Obesity develops when caloric intake exceeds energy expenditure. Thus, it is important to study the underlying mechanisms in the regulation of energy metabolism.

Adipose tissue is a key organ in the regulation of energy homeostasis, as it hosts three kinds of adipocytes with distinct features: white adipocytes, brown adipocytes and beige adipocytes. White adipose tissue (WAT), which is equipped with metabolic pathways that can efficiently convert excess energy into triglyceride, is the main organ for energy storage in the body and is the determinant of the overall adiposity [2]. WAT mass can expand via hypertrophy or hyperplasia or a combination of both. While adipocyte hypertrophy results from the uptake of extracellular lipids by lipoproteins or production of intracellular lipids by de novo lipogenesis, hyperplasia involves an adipogenic process in which adipose progenitor cells (APCs) undergo proliferation and differentiation to form mature adipocytes [3]. Unlike WAT, mouse brown adipose tissue (BAT), which is mainly localized to anatomical regions such as the interscapular region, functions to dissipate energy via heat production through both UCP1-dependent and UCP1-independent mechanisms [4,5,6,7]. Mainly induced by β-adrenergic activation via cold or β-adrenergic agonists, beige adipocytes are dispersed within WAT depots. Beige fat also possesses the thermogenic properties and shares some common morphological and biochemical features with classic BAT [8]. All three adipocytes, whose development and function are tightly regulated by nutritional and hormonal signaling, collectively play a significant and yet distinct role in maintaining energy homeostasis.

Obesity is a polygenic disease involving both genetics and environmental factors, the interplay of which is mediated by epigenetic pathways [9,10]. We have identified several epigenetic factors mediating white fat adipogenesis and brown fat thermogenesis [11,12,13,14]. We previously reported that ubiquitously transcribed tetratricopeptide (*Utx*), a histone demethylase that specifically demethylates histone 3 lysine 27 di- or tri-methylation (H3K27me2 or H3K27me3) [15], promotes brown adipocyte thermogenic gene expression such as *Pgc1α* and *Ucp1* via a dynamic epigenetic event involving H3K27me3 demethylation followed by H3K27 acetylation [14]. Our very recent study further determined the physiological significance of *Utx* in vivo using a brown fat *Utx* knockout mouse and discovered that *Utx* deficiency in BAT impairs cold-induced thermogenesis, reduces energy expenditure and exacerbates diet-induced obesity [16]. To further determine the role of *Utx* in overall adipose tissue energy metabolism, we here created a new mouse model with *Utx* deficiency in mature adipocytes (AUTXKO mice) and characterized the metabolic phenotypes of this model. We discovered an unexpected role of adipocyte *Utx* in the development of white adipose tissue remodeling and dysfunction and hepatic steatosis under HFD-induced metabolic stress.

## 2. Materials and Methods

### 2.1. Animal Models

Animal models with adipocyte-specific deletion of *Utx* (AUTXKO) were generated by breeding *Utx*-floxed mice [17] (Jackson Lab, stock No. 021926) with Adiponectin-*Cre* mice (Jackson Laboratory, Stock No 028020), in which *Cre* is specifically expressed in mature fat cells including white and brown fat cells. *Utx* is localized on the X chromosome [18]. Thus, male UTXKO animals were designated as *Utx^fl/Y^* with Adiponectin-*Cre*, with *Utx^fl/Y^* littermates as controls [16]. Animals used in the study were maintained in a pathogen-free facility with a 12-h light/dark cycle and an ambient temperature at 20–22 °C. All the procedures performed in the animal studies were approved by the Institutional Animal Care and Use Committee at Georgia State University.

### 2.2. Characterization of Metabolic Phenotype

AUTXKO and their respective littermate control animals were weaned on a standard chow diet (LabDiet 5001, LabDiet, St. Louis, MO, USA) and were maintained on the same diet until the end of the study after 36 weeks. Another cohort of AUTXKO mice and the control mice were challenged with a high-fat diet (HFD) (Research Diets D12492, 60% calorie from fat) for 10 weeks. Various metabolic measurements were conducted. Body weight of mice was weighed weekly throughout the experiments. Animal food intake was monitored in metabolic cages through a period of 5 days. Body composition of mice was determined by a Minispec LF90II NMR body composition analyzer (Bruker BioSpin Corporation; Billerica, MA, USA). Indirect calorimetry with PhenoMaster Automatic Homecage Phenotyping system (TSE Systems, Chesterfield, MO, USA) was conducted to assess energy expenditure and physical activity. A drop of blood from tail nick was used to measure glucose levels using a portable Glucose meter (OneTouch Ultra, LifeScan, Milpitas, CA, USA). A glucose tolerance test (GTT) and insulin tolerance test (ITT) were conducted to assess glucose tolerance and insulin sensitivity in mice [19,20]. At the termination of the experiments, various organs, including fat and liver, were dissected, weighed and snap-frozen for the assays described below.

### 2.3. Body Temperature Determination

A temperature transponder (BioMedic Data Systems, Seaford, DE, USA) was surgically embedded in the abdominal cavity of male AUTXKO mice and their *fl/Y* littermates for monitoring core temperature. After a one-week recovery, animals underwent a chronic seven-day cold exposure at 5–6 °C [21]. 

### 2.4. Real-Time RT-PCR

Snap-frozen tissues were homogenized in Tri Reagents (Molecular Research Center, Cincinnati, OH, USA) [22] to isolate total RNA. RNA levels of genes-of-interest were quantified by a real-time RT-PCR approach using TaqMan Universal PCR Master Mix (ThermoFisher Scientific, Waltham, MA, USA) on a QuantStudio 3 real-time PCR machine (ThermoFisher Scientific) [22]. The primer-probe pairs for the genes measured were obtained from Applied Biosystems (ThermoFisher Scientific).

### 2.5. Western Blot

Western blot was conducted to assess the protein of interest [23]. Snap-frozen samples were homogenized in a modified radioimmunoprecipitation assay buffer and followed by centrifugation to extract supernatants. An equal amount of protein was loaded on SDS-PAGE, which was transferred to nitrocellulose membranes (Bio-Rad, Hercules, CA, USA). The proteins were then immunoblotted with various primary and secondary antibodies, and visualized on a Li-COR Imager System (Li-COR Biosciences, Lincoln, NE, USA). The protein antibodies are listed as follows: UCP1 antibody (1:500, abcam, ab23841), Mitochondrial total OXPHOS protein antibody set (Abcam, ab110413), phospho-HSL antibody (pHSL) (1:1000, Cell Signaling 4126), HSL antibody (1:1000, Cell Signaling 4107), α-Tubulin antibody (1:1000, Advanced BioChemicals, ABCENT4777, Lawrenceville, GA, USA) and the secondary antibody Alexa Fluor 680 (ThermoFisher Scientific).

### 2.6. Immunohistochemistry (IHC)

Fat samples were fixed in formalin, paraffin-embedded and manually sliced into 5–6 µm sections. The sections were stained with hematoxylin and eosin (H&E), or incubated with the anti-UCP1 antibody (1:150, abcam, ab10983), which was then developed with DAB peroxidase substrate (Vector Labs, SK-4100) [24,25]. 

### 2.7. RNA-Sequencing

Total RNA was extracted from epididymal WAT and was sent to Beijing Genomics Institute (BGI, Shenzhen, China) for sequencing [20]. Clean reads were submitted to UCSC genome browser mouse mm10 for alignment using TopHat. Genes were considered to have a differential expression with a difference set at a Log2 fold change ≥0.5 or ≤−0.5, and FDR < 0.001.

### 2.8. Statistical Analysis

Data were presented as mean ± S.E.M. Statistical difference among groups was analyzed by one-way ANOVA or t-test as appropriate using GraphPad Prism 5. *p* < 0.05 was considered as statistically significant.

## 3. Results

### 3.1. Utx Deficiency in Adipocytes Increases Adiposity in Mice Fed a Chow Diet 

Our recent findings demonstrated the importance of *Utx* in regulating brown fat thermogenic function and diet-induced obesity [14,16]. To gain a better understanding of *Utx* in overall adipose tissue biology, we created a mouse with *Utx* deletion specifically in adipocytes (AUTXKO) by breeding the *Utx*-floxed mouse with the Adiponectin-*Cre* line. *Utx* mRNA in inguinal white adipose tissue (iWAT), epididymal WAT (eWAT) and interscapular brown adipose tissue (iBAT) of the knockout mice was reduced by 70%, 70% and 50%, respectively (Appendix A). We first conducted the metabolic phenotyping on male AUTXKO mice maintained on a standard chow diet. AUTXKO animals did not show any difference in body weight until they were 30 weeks old, when they started to display a heavier body weight, suggesting an aging-dependent development of obesity (Figure 1A). We also found increased fat mass with concomitantly decreased lean mass in male AUTXKO animals compared to their *fl/Y* controls (Figure 1B). Moreover, AUTXKO mice also exhibited increased organ weight in fat depots, including iBAT, iWAT, eWAT and liver (Figure 1C). In addition, AUTXKO animals exhibited a tendency towards a lower oxygen consumption, particularly during the dark cycle, albeit not reaching statistical significance (Figure 1D), without changes in respiratory exchange ratio (RER), activity and food consumption (Figure 1E–G). Since we previously reported that *Utx* positively regulates BAT thermogenesis [14,16], we sought to examine BAT function in AUTXKO mice where the *Utx* expression is also reduced due to the presence of Adiponectin-driven Cre. We found that UCP1 protein level was decreased in iBAT of AUTXKO mice, which was associated with a down-regulation of mitochondrial respiratory chain proteins, including mitochondrial ATP synthase F1 subunit alpha (ATP5F1A) in complex V and mitochondrial Cytochrome b-c1 complex subunit 2 (UQCRC2) in complex III (Figure 2). We also examined the mitochondrial protein, UCP1 protein and phosphor-hormone sensitive lipase (HSL) in iWAT. As expected, UCP1 protein bands in iWAT of both AUTXKO and *fl/Y* mice were hardly detectable compared to that of iBAT (Appendix A), which serves as a positive control for UCP1 protein detection. No differences were found in mitochondrial protein and phosphor-HSL between the two genotypes (Appendix A). These data suggest that the down-regulation of UCP1 protein and the mitochondrial program in brown fat by *Utx* deficiency may contribute to the tendency of decreased energy expenditure and increased adiposity in AUTXKO mice.

Since insulin sensitivity is associated with adiposity levels, we next conducted a glucose tolerance test (GTT) and insulin tolerance test (ITT) in AUTXKO mice. As AUTXKO animals had higher fat mass, they also displayed impaired insulin sensitivity in response to a glucose or insulin bolus challenge (Figure 3A,B). 

### 3.2. Utx Deficiency in Adipocytes Impairs Cold-Induced Thermogenesis

We next characterized the cold-induced thermogenesis in AUTXKO animals by subjecting them to a chronic cold for 7 days. AUTXKO animals exhibited a cold intolerance evident by a lower core temperature during the cold exposure (Figure 4A), suggesting a cold intolerance in the knockout mice. Associated with a reduced body temperature, protein expression of UCP1 and the mitochondrial respiratory chain complexes were also down-regulated in iBAT of AUTXKO animals, including ATP5F1A, succinate dehydrogenase complex subunit B (SDHB) in complex II and NADH dehydrogenase 1β subcomplex 8 (NDUFB8) in complex I (Figure 4B). Further immunohistochemical assessment revealed a lesser UCP1 protein in the brown fat of AUTXKO animals (Figure 4C), which, along with decreased mitochondrial respiratory chain protein, may explain the impaired cold-induced thermogenesis in AUTXKO mice.

### 3.3. Utx Deficiency in Adipocytes Causes High-Fat Diet-Induced Metabolic Dysfunction

We next put AUTXKO mice on HFD and measured their metabolic phenotypes. Figure 5A shows that AUTXKO mice gained significantly more weight on HFD. This was congruent with an increase in body fat composition of AUTXKO animals in comparison to their controls (Figure 5B). Surprisingly, we found a decrease in the weight of WAT depots, including iWAT, eWAT and retroperitoneal WAT (rWAT) in AUTXKO animals (Figure 5C). By contrast, AUTXKO mice exhibited significantly increased liver weight (Figure 5C), suggesting that the fat-laden liver, instead of WAT, might be the major reservoir for lipid storage and the major contributor to the increased fat composition and body weight of AUTXKO mice. In addition, we did not find any differences in oxygen consumption, RER, locomotor activity and food consumption between AUTXKO animals and the controls (Figure 5D–G). To determine whether brown and beige fat thermogenesis is impaired by adipocyte *Utx* deficiency, we examined the mitochondrial respiratory chain protein, UCP1 protein and phosphor-HSL in iBAT and iWAT. We did not find any differences in UCP1 protein and phosphor-HSL between the two genotype groups, although there was a decrease in ATP5F1A and UQCRC2 protein in the iBAT of AUTXKO mice (Appendix A). We also did not observe any changes in UCP1 protein, phosphor-HSL and mitochondrial respiratory chain protein in iWAT of AUTXKO mice compared to the controls (Appendix A). These data suggest that brown and beige fat thermogenesis may not be a key factor contributing to the reduced adipose tissue mass in AUTXKO mice.

We further characterized glucose metabolism in these animals. A GTT experiment showed a higher glycemic response to intraperitoneal injection of glucose in AUTXKO animals in comparison to *fl/Y* animals (Figure 6A), while an ITT experiment revealed a blunted hypoglycemic response to intraperitoneal injection of insulin in the knockout mice (Figure 6B), suggesting that Utx deficiency in adipocytes causes glucose intolerance and insulin resistance.

To investigate the pathways underlying the shift of lipid storage from WAT to the liver, we examined the gene expression profiles of WAT in AUTXKO and *fl/Y* mice. We found that *Utx*-deficient WAT displayed a down-regulation of genes in adipogenesis and adipocyte phenotype, including peroxisome proliferator activated receptor gamma (*Pparγ*), sterol regulatory element binding protein 1C (*Srebp1c*), stearoyl-Coenzyme A desaturase 1 (Scd1), adiponectin, C1Q and collagen domain containing (*Adipoq*) and an up-regulation of genes in inflammation, macrophage content and fibrosis, including nitric oxide synthase 2, inducible (*Nos2*), tumor necrosis factor (*Tnfα*), interleukin 1β (*Il-1β*), chemokine (C-C motif) receptor 7 (*Ccr7*), transforming growth factor β1 (Tgfβ1) and collagen, type I, alpha 2 (*Col1a2*) (Figure 7A), suggesting a WAT remodeling featured by increased inflammation and fibrogenesis. Further RNA-seq analysis revealed an increase in pathways regulating inflammation and fibrosis in the eWAT of AUTXKO mice (Figure 7B). In consistence, immunohistochemical staining of CD68 indicated more macrophages in the eWAT of AUTXKO animals compared to the controls (Figure 7C). 

We next conducted metabolic characterization on the liver, which exhibited a higher weight in the knockout mice. Histological analysis revealed more lipid accumulation in the hepatocytes of AUTXKO mice (Figure 8A), suggesting a more severe steatosis in the knockout mice. Moreover, the increased hepatic steatosis of AUTXKO mice was associated with increased triglyceride levels in circulation (Figure 8B). Real-time RT-PCR measurements showed an up-regulation of genes responsible for inflammatory responses, including nitric oxide synthase 1, neuronal (*Nos1*), interleukin 6 (*Il-6*) and chemokine (C-C motif) ligand 2 (*Ccl2*), and fibrosis, such as collagen, type III, alpha 1 (*Col3a1*) (Figure 8C). Immunohistochemical analysis revealed more macrophages stained by CD68 antibody in the liver of AUTXKO animals (Figure 8D). Our study suggests that adipocyte *Utx* deficiency causes metabolic dysfunction that limits fat accumulation in adipose tissue and increases lipid overflow to the liver for storage.

## 4. Discussion

In the current studies, we demonstrated the importance of adipocyte *Utx* in overall energy metabolism and glucose homeostasis. *Utx* deficiency in adipocytes causes metabolic dysfunction in both WAT and liver. The scientific premise of this study was based on our previous studies on brown fat *Utx*. Firstly, we reported that *Utx* positively regulates brown adipocyte thermogenic gene expression in vitro [14]. Secondly, we recently examined the physiological significance of brown fat *Utx* in vivo and discovered that *Utx* deficiency in BAT impairs cold-induced thermogenesis, decreases energy metabolism and promotes diet-induced obesity [16]. This might be attributed to a brown fat remodeling featured by a dramatic up-regulation of myogenic markers in the BAT of the knockout mice. The conversion of functional brown adipocytes into myocyte-like brown fat compromised oxygen consumption rate, presumably contributing to reduced energy expenditure in the knockout mice [16]. In consistent with our prior discovery, AUTXKO mice, at least on chow diet, appear to have a lower oxygen consumption, because *Utx* deletion also takes place at the iBAT of AUTXKO mice due to the expression of Adiponectin-driven *Cre*. However, HFD-challenged AUTXKO mice present a phenotype that mainly lies within WAT including enhanced inflammation, fibrosis and less capacity to store lipids, which differs from that of brown fat *Utx* knockout mice. Although the mechanism underlying the WAT remodeling due to *Utx* deficiency is not entirely clear and warrants a further investigation, dysfunctional adipose tissue manifested by inflammation and fibrosis has a detrimental impact on whole-body metabolic homeostasis [26,27]. For instance, inflammation and fibrogenesis induced by hypoxia have been shown to limit a healthy expansion of adipose tissue [26,27], which resembles the phenotype of AUTXKO mice. The direct consequences of the adipose dysfunction include the production of exaggerated inflammatory mediators that pose detriments to other metabolic organs and an increase of lipid spillover to other non-adipose organs for storage that causes lipotoxicity, a double assault that ultimately leads to the metabolic disorders often seen in obesity [26,27]. By contrast, mice with an ability to expand adipose tissue, such as the adiponectin-overexpressing model, display an improved metabolic profile during obesity development [28]. It is noteworthy that a diet-induced adipose dysfunction was also observed in AUTXKO mice. Upon HFD challenge, AUTXKO mice are not able to handle excess nutrient influx into adipose tissue for a proper storage. The reduced fat mass in adipose tissues likely stems from the inability of adipocytes to store lipids evident by a decreased lipogenic program and a potentially increased lipolysis due to a tendency of increased phosphor-HSL. The defect in lipid handling is further exacerbated by increased inflammation and fibrosis, a hallmark of adipose tissue remodeling and dysfunction. The impaired lipid metabolism coupled with increased inflammation and fibrosis limits lipid accumulation in adipose tissue of AUTXKO mice, which in turn elicits lipid overflow into the liver to cause steatosis and could potentially develop into nonalcoholic steatohepatitis (NASH) featured with inflammation, macrophage infiltration and fibrosis. Exaggerated adipose inflammation coupled with hepatic steatosis would lead to obesity-induced metabolic dysfunction, such as insulin resistance, as we observed in AUTXKO mice. Nonetheless, future studies involving a careful examination of circulating fatty acids, fatty acid transport in the liver and lipoprotein lipase activity will be warranted to depict the exact pathways underlying the shift of lipid storage from adipose tissue to the liver in AUTXKO mice.

While AUTXKO mice on a regular chow diet exhibit an obese phenotype, these mice have reduced fat mass when challenged with HFD. The exact reason for the discrepancy in the phenotypes between chow and HFD-fed AUTXKO mice is not clear. One possibility would be that *Utx* may differentially regulate adipocyte lipid homeostasis depending on the dietary and nutritional conditions. We discovered that *Utx*-deficient WAT displayed a down-regulation of genes in lipogenesis and adipocyte phenotype, including *Pparγ*, *Srebp1c*, and *Scd1* in HFD-fed mice. Consistently, a recent study reported deleting *Utx* in adipocytes by the aP2-Cre line reduces fat mass and prevents diet-induced obesity [29]. This genetic model with adipocyte *Utx* deficiency however displays reduced fatty liver. Of note, the aP2-Cre line has a Cre expression in adipocytes as well as macrophages, which inevitably deletes *Utx* gene in macrophages. Thus, macrophage deletion of *Utx* may confound the phenotype by altering macrophage inflammation and lipid metabolism in both adipocytes and the liver through a paracrine action due to the infiltration of macrophages in both tissues [30]. Indeed, *Utx* deficiency in macrophage has been shown to regulate macrophage polarization and promote brown fat thermogenesis [31], which may alter both systemic and hepatic lipid metabolism. Nonetheless, the use of different Cre lines to generate tissue specific *Utx* knockout may explain the discordant phenotypes in hepatic steatosis between our study and the one reported by Ota et al. [29].

To summarize our findings, we find adipocyte *Utx* deficiency increases fat mass and body weight, leading to insulin resistance in mice fed a chow diet. When challenged with HFD, mice with adipocyte Utx deficiency exhibit adipose dysfunction featured by exacerbated inflammation and fibrosis with less fat storage in adipose tissues and more lipid accumulation in the liver; as a result, the mice display glucose intolerance and insulin resistance. Our study supports the importance of adipocyte *Utx* in regulating lipid homeostasis and partitioning between adipose tissue and liver during obesity development. 

## Figures and Tables

**Figure 1 cells-11-00181-f001:**
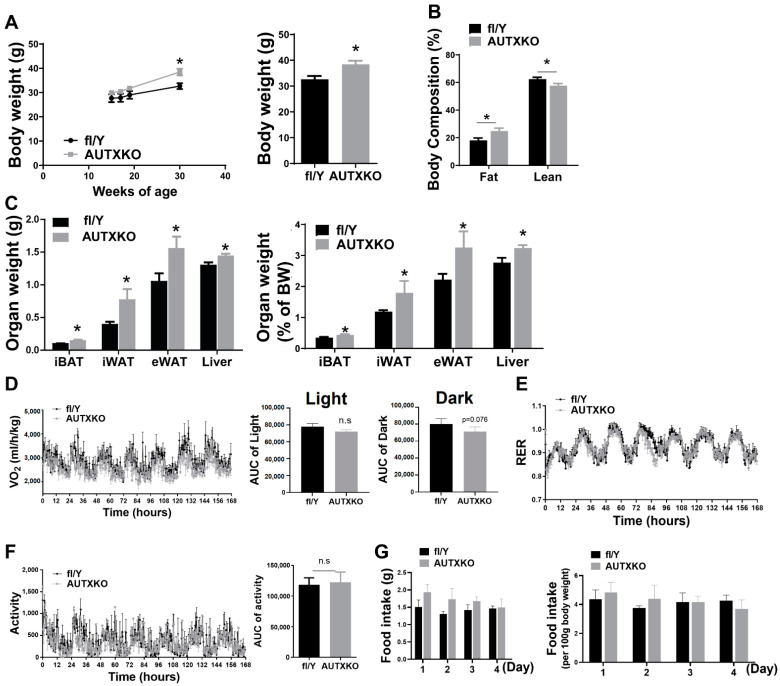
*Utx* deficiency in adipocytes reduces adiposity in mice fed a chow diet. Three-week-old male AUTXKO and their littermate *fl/Y* controls were maintained on a regular chow diet throughout the experiment. (**A**–**C**): (**A**) Body weight in male AUTXKO and *fl/Y* animals in a time course (**left**) and at the age of 30 weeks (**right**). (**B**) Fat and lean mass composition in male AUTXKO and *fl/Y* animals at the age of 30 weeks. (**C**) Absolute organ weight (**left**) and organ weight normalized to body weight (**right**) of iBAT, iWAT, eWAT and liver in male AUTXKO and *fl/Y* animals at the age of 36 weeks. (**D**–**G**): 33-week-old male AUTXKO and *fl/Y* animals maintained on a chow diet were put in TSE PhenoMaster metabolic cage system for metabolic characterization, including: (**D**) Oxygen consumption; (**E**) Respiratory exchange ratio (RER); (**F**) Locomotor activity; (**G**) Daily food consumption presented as food gram/mouse/day (**left**) and food gram/100 g body weight (**right**). All data are presented as mean ± S.E.M.; *n* = 4/group; * *p* < 0.05 vs. *fl/Y*.

**Figure 2 cells-11-00181-f002:**
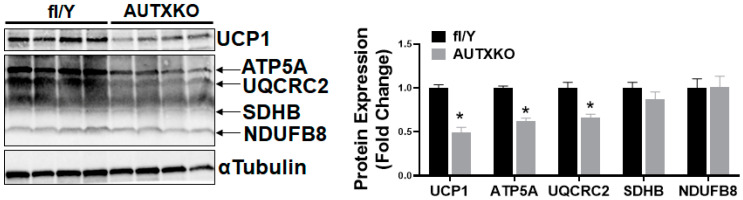
*Utx* deficiency impairs brown fat thermogenic program in mice fed a chow diet. Three-week-old male AUTXKO and *fl/Y* animals were maintained on a regular chow diet throughout the experiment. UCP1 protein and mitochondrial respiratory chain complex protein in iBAT were assessed by western blot. Left: representative blot images; right: densitometry quantitation. All data are presented as mean ± S.E.M.; *n* = 4/group; * *p* < 0.05 vs. *fl/Y*.

**Figure 3 cells-11-00181-f003:**
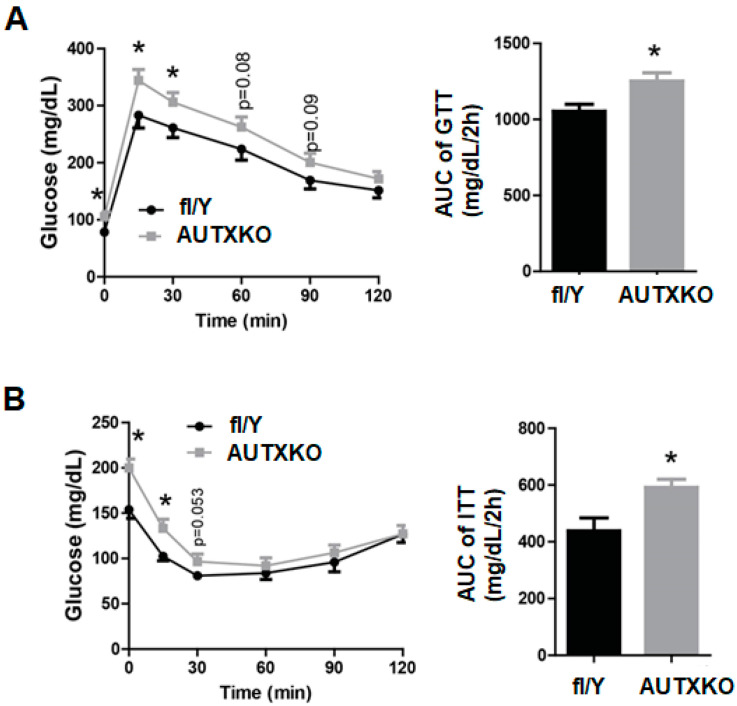
*Utx* deficiency in adipocytes causes glucose intolerance in mice fed a chow diet. Three-week-old male AUTXKO and *fl/Y* animals were maintained on a regular chow diet throughout the experiment. (**A**) GTT performed in 31-week-old male AUTXKO and *fl/Y* animals. (**B**) ITT performed in 32-week-old male AUTXKO and *fl/Y* animals. All data are presented as mean ± S.E.M.; *n* = 8/group; * *p* < 0.05 vs. *fl/Y*.

**Figure 4 cells-11-00181-f004:**
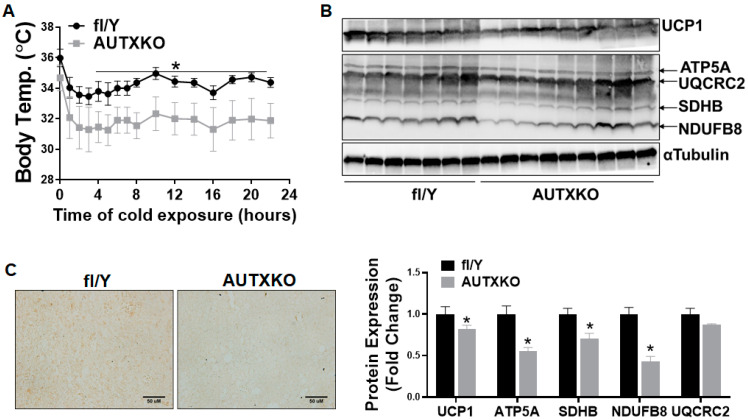
*Utx* deficiency in adipocytes reduces brown fat thermogenesis in cold-challenged animals. Male AUTXKO and *fl/Y* animals at 5 months of age underwent a 7-day cold challenge at 5 °C. (**A**) Body temperature of male AUTXKO and *fl/Y* animals. (**B**) Western blots of UCP1 and mitochondrial electron transport chain proteins in the brown fat of male AUTXKO and *fl/Y* animals with blot images on top the panel and quantitation of blot densitometry on the bottom panel. (**C**) Immunohistochemical (IHC) staining of UCP1 in the brown fat of male AUTXKO and *fl/Y* animals. All data are presented as mean ± S.E.M.; *n* = 7–9/group; * *p* < 0.05 vs. *fl/Y*.

**Figure 5 cells-11-00181-f005:**
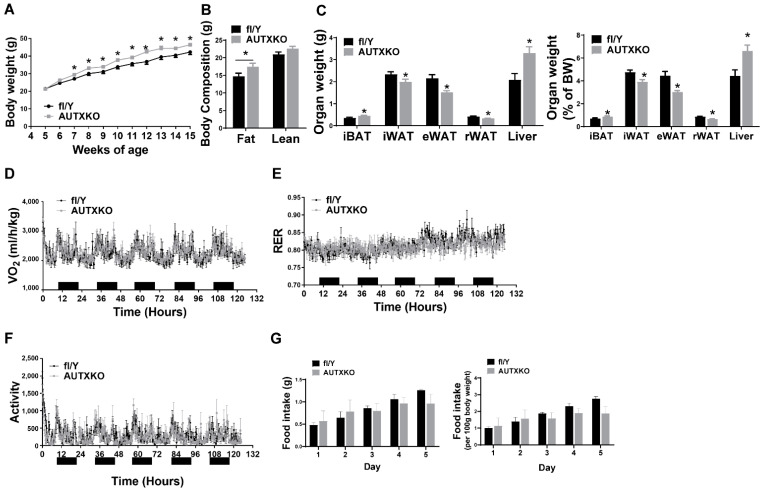
*Utx* deficiency in adipocytes reduces fat mass while increasing liver weight in mice fed a high-fat diet. Five-week-old male AUTXKO and *fl/Y* animals were challenged with an HFD. (**A**–**C**): (**A**) Body weight; (**B**) Fat and lean mass composition; (**C**) Organ weight (**left**) and organ weight normalized to body weight (**right**) of iBAT, iWAT, eWAT, rWAT and liver in male AUTXKO and *fl/Y* animals. (**D**–**G**): 13-week-old male AUTXKO and *fl/Y* mice fed HFD were put in TSE PhenoMaster metabolic cage system for metabolic characterization. (**D**) Oxygen consumption; (**E**) RER; (**F**) Locomotor activity; (**G**) Daily food consumption presented as food gram/mouse/day (**left**) and food gram/100 g body weight (**right**) in HFD-fed AUTXKO and *fl/Y* animals. All data are presented as mean ± S.E.M.; *n* = 4/group; * *p* < 0.05 vs. *fl/Y*.

**Figure 6 cells-11-00181-f006:**
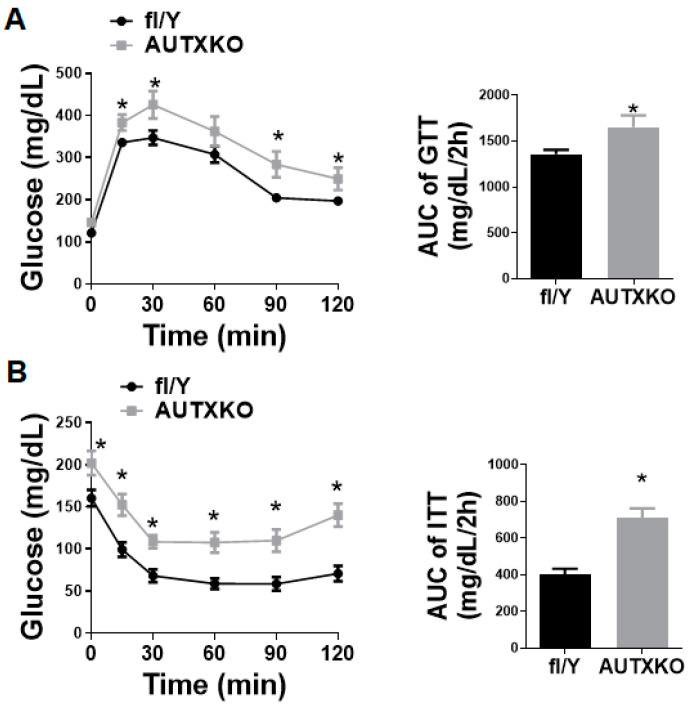
*Utx* deficiency in adipocytes impairs insulin sensitivity in mice fed HFD. Five-week-old male AUTXKO and *fl/Y* animals were challenged with an HFD. (**A**) GTT in 16-week-old male AUTXKO and *fl/Y* animals. (**B**) ITT in 17-week-old male AUTXKO and *fl/Y* animals. All data are presented as mean ± S.E.M.; *n* = 11/group; * *p* < 0.05 vs. *fl/Y*.

**Figure 7 cells-11-00181-f007:**
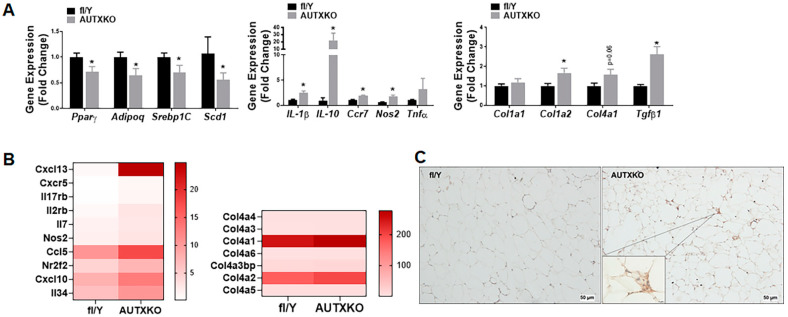
*Utx* deficiency in adipocytes causes metabolic dysfunction in adipose tissue. Five-week-old male AUTXKO and *fl/Y* animals were challenged with an HFD. (**A**) RT-PCR measurements of genes encoding lipogenesis, inflammation and fibrosis in WAT. All data are presented as mean ± SEM; *n* = 7–11/group; * *p* < 0.05 vs. *fl/Y*. (**B**) RNA-seq analysis reveals an up-regulation of pathways underlying inflammation and fibrosis in the eWAT of AUTXKO mice. (**C**) Immunohistochemical staining of macrophages with an anti-CD68 antibody in the white adipose tissue.

**Figure 8 cells-11-00181-f008:**
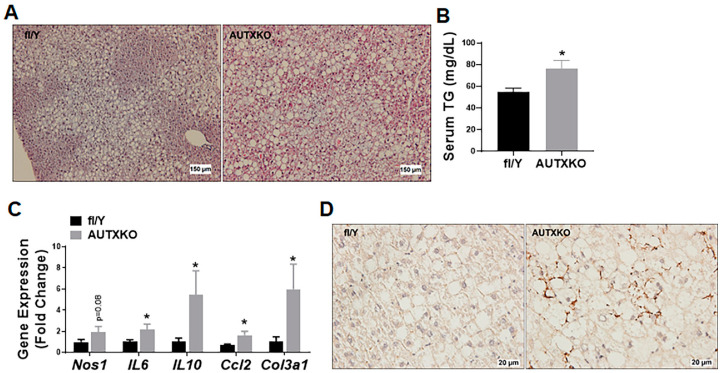
*Utx* deficiency in adipocytes causes hepatic steatosis in AUTXKO mice. Five-week-old male AUTXKO and *fl/Y* animals were challenged with an HFD. (**A**) H&E staining of the liver. (**B**) Triglyceride levels in circulation. (**C**) RT-PCR measurements of genes encoding inflammatory and fibrogenic pathways in the liver of AUTXKO and the control mice. (**D**) Immunohistochemical analysis of macrophages stained with an anti-CD68 antibody in the liver. All data are presented as mean ± S.E.M.; *n* = 7–11/group; * *p* < 0.05 vs. *fl/Y*.

## Data Availability

All datasets will be available upon request to the corresponding authors Hang Shi and Bingzhong Xue.

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
