# Peer review of "Adipocyte Utx Deficiency Promotes High-Fat Diet-Induced Metabolic Dysfunction in Mice"

_cells, 2022, doi:10.3390/cells11020181_

Round 1

Reviewer 1 Report

In this study, Li et al demonstrated that adipose Utx depletion led to increased weight and fat mass gain and improved glucose tolerance and insulin resistance in mice fed with both chow and high-fat (HFD) diet. Mechanistically, they suggest that the obesogenic effect of adipose Utx depletion might be through the inhibition of thermogenic activity in iBAT, decreasing, in consequence, the energy expenditure. The problem is that the thermogenic activity in iBAT (UCP1 western blot) was only analysed in chow diet. Otherwhise, in HFD, no differences on energy expenditure were found.

Then, the possible mechanism to explain the obesogenic effect of adipose Utx depletion remains open.

Major points:

-Thermogenic and mitochondrial proteins also should be analysed in WAT (both in inguinal and visceral fat), and not only in iBAT, in mice fed with chow diet.

-The inhibitory effect of Utx depletion on BAT activity and obesity is not novel. It has been recently published by the same authors (PMID: 34824202), as they mentioned in the manuscript.

-Thermogenic and mitochondrial proteins should be analysed in WAT and iBAT in mice fed with HFD.

-Lipolytic enzymes also should be analysed in both WAT and iBAT.

-The contradictions with the studies: PMID: 33303977, PMID: 31296899 are disturbing.

Author Response

“In this study, Li et al demonstrated that adipose Utx depletion led to increased weight and fat mass gain and improved glucose tolerance and insulin resistance in mice fed with both chow and high-fat (HFD) diet. Mechanistically, they suggest that the obesogenic effect of adipose Utx depletion might be through the inhibition of thermogenic activity in iBAT, decreasing, in consequence, the energy expenditure. The problem is that the thermogenic activity in iBAT (UCP1 western blot) was only analysed in chow diet. Otherwhise, in HFD, no differences on energy expenditure were found.” “Then, the possible mechanism to explain the obesogenic effect of adipose Utx depletion remains open.”

Thank you Reviewer 1 for the comments on the obesogenic effect of adipocyte Utx in chow and HFD-fed mice. Although the exact mechanism is not clear, two different pathways in AUTXKO mice are altered by the chow and HFD feeding respectively, resulting in a discordant phenotype between chow-fed and HFD-fed AUTXKO mice. AUTXKO mice on chow diet exhibit increased body weight and fat mass, which may result from a decreased energy expenditure due to impaired thermogenic program in iBAT. Conversely, when challenged with HFD, AUTXKO mice display a reduced adiposity evident by a lower fat mass in fat depots albeit a shift of lipid storage from WAT to liver. The reduced fat mass in adipose tissues likely stems from the inability of adipocytes to store lipids evident by a decreased lipogenic program, which is further exacerbated by increased inflammation and fibrosis, a hallmark of adipose tissue remodeling and dysfunction. Indeed, the energy expenditure in HFD-fed AUTXKO mice did not change, nor did iBAT UCP1 protein in iBAT of (except a decrease in two out of four mitochondrial protein) the knockout mice. The latter was presented as a new figure in Supplemental Figure 3. In sum, the reduced adiposity in AUTXKO mice fed HFD is mainly caused by the inhibition of lipogenesis in WAT while the obese phenotype of chow-fed AUTXKO mice may originate from impaired thermogenic program in BAT.

Major points:

-Thermogenic and mitochondrial proteins also should be analysed in WAT (both in inguinal and visceral fat), and not only in iBAT, in mice fed with chow diet.

Thanks for the comment. Beige adipocytes in WAT depots of adult mice are mainly induced by cold or β-adrenergic agonists and are not commonly seen in WAT of mice housed at room temperature. Hence UCP1 protein as a marker for beige adipocytes is barely detectable at room temperature in iWAT, the fat depot harboring most beige cells, and is almost non-existing in visceral WAT. Since Utx deficiency is predicted to further reduce UCP1 protein as it does to iBAT, experiments as suggested are difficult to assess the change of UCP1 protein due to lack of a proper baseline level of UCP1 for comparison. Nonetheless, we did conduct an immunoblotting of UCP1 and mitochondrial protein using iWAT of mice housed at room temperature. As expected, UCP1 protein bands in iWAT of the control fl/Y mice are hardly detectable compared to that of iBAT, which serves as a positive control for UCP1 protein detection. No UCP1 protein bands are detectable in iWAT of AUTXKO mice either. The data are presented in Supplemental Figure 2.

-The inhibitory effect of Utx depletion on BAT activity and obesity is not novel. It has been recently published by the same authors (PMID: 34824202), as they mentioned in the manuscript.

We indeed reported the role of Utx in regulation of brown fat thermogenesis. However, in the published study we employed a genetic model with Utx deletion in brown adipocytes by crossing the Utx fl/fl line with the Ucp1-Cre line. Our current model with deletion of Utx in all adipocytes including brown and white fat by using the Adiponectin Cre line is different from the one published. More importantly, the present study using the new model uncovered a novel role of Utx in the development of diet-induced adipose tissue dysfunction, which has never been reported before by us or others. In addition, our present study further confirmed the finding on the role of brown fat Utx in brown fat thermogenesis, which underscores the reproducibility of our prior study, an area of focus emphasized by the NIH sponsored research.

“Thermogenic and mitochondrial proteins should be analysed in WAT and iBAT in mice fed with HFD.”

Thanks for the comments.

  1. As we stated above, beige cells do not normally develop in WAT of mice fed high fat diet and hence UCP1 protein in iWAT remains physiologically scarce. Nonetheless, we did conduct an immunoblotting of UCP1 and mitochondrial protein using iWAT of mice fed HFD. As expected, UCP1 protein expression is quite low in iWAT of both AUTXKO mice and the control fl/Y mice. Compared to the UCP1 protein level in BAT (a positive control in the blot of Suppl. Fig. 4), UCP1 protein in WAT at such a low level is physiologically insignificant. The data were presented in Supplemental Figure
  2. We also conducted immunoblotting of UCP1, phosphor-HSL and mitochondrial respiratory chain complex protein using iBAT of mice fed HFD. No differences were found in UCP1 and phosphor-HSL between the two genotypes, although there was a reduction in two out of four mitochondrial protein in AUTXKO mice. The data were presented in Supplemental Figure 3.

“Lipolytic enzymes also should be analysed in both WAT and iBAT.”

We conducted an immunoblotting of phosphor-HSL, a marker of lipolytic activation, in iWAT and iBAT of AUTXKO mice and their controls. When normalized to total HSL, there was a tendency (but not statistically significant) towards increase of phosphor-HSL in iWAT of AUTXKO mice and the results were presented in Supplemental Figure 2 and 4.

“The contradictions with the studies: PMID: 33303977, PMID: 31296899 are disturbing.”

Thanks for the comment. However, we were fully aware of these two papers during the course of our studies and preparation of our manuscript so that we cited these two papers and discussed them at length in the section of discussion on page 10, line 347-358. The discrepancy of the phenotypes between our model AUTXKO and the ones reported in the two papers can be well accounted for by the use of the different genetic models in the respective study.

  1. The study PMID: 33303977 used a model with myeloid deletion of Utx, which is completely different from our AUTXKO model with the deletion of Utx in adipocytes. It is conceivable that adipocyte Utx knockout mice and macrophage Utx knockout mice may not have the same phenotypes because like many other genes, Utx may play a distinct role in regulating physiological functions in adipocytes and macrophages respectively.
  2. The other study PMID: 31296899 characterized a model with deletion of Utx in adipocytes by the aP2-Cre line, which exhibits a phenotype that is somewhat congruent with ours, despite use of a Cre line different from ours. Both models exhibit reduced fat mass when challenged with HFD. One caveat is the hepatic lipid accumulation that is different between the two models. Our AUTXKO mice develop hepatic steatosis that was absent from theirs. It should be noted that employing the aP2 Cre line to delete a gene of interest is not as clean as the use of Adiponectin Cre line because the former has a Cre leaking in macrophages, which inevitably deletes Utx gene in macrophages. It is possible that Utx deficiency in macrophages may confound the phenotype by altering macrophage inflammation and lipid metabolism in both adipocytes and liver through a paracrine action due to the infiltration of macrophages in both tissues. As we discussed in point A) above, Utx deficiency in macrophage has been shown to regulate macrophage polarization and promote brown fat thermo-genesis, which may alter both systemic and hepatic lipid metabolism. In sum, the use of different Cre lines to generate tissue specific Utx knockout may explain the discordant phenotypes in hepatic steatosis between our study and the one report-ed by Ota et al.

Reviewer 2 Report

This work adds another important facet of the understanding surrounding the field of adipocyte biology. That Utx is able to alter the expression of genes important for mitochondrial function and UCP1 by acting as a histone demethylase is interesting. This is similar to a switch that activates brown adipocytes and facilitates the browning process in WAT. The authors have elegantly demonstrated that selective knockout of Utx in adipocytes impair the capacity of WAT to store fat and also diminishes BAT activity. The result is an increased in ectopic fat deposition especially in the liver, accompanied by elevated insulin resistance, insulin response )seen in the ITT curves) and impaired glucose tolerance (seen in the GTT curves). 

I think a brief description of how Utx alters chromatin structure and enhances the transcription of genes relevant to adipocyte biology would be good in the Introduction. 

A few points I wish to bring up for the authors to comment:

  1.  For the assessment of BAT activity, could they have employed PET-CT scanning or infrared thermography or even fat fraction MRI to determine the activation of BAT in vivo in real time during cold exposure?
  2. Was the thyroid function (ie. plasma free thyroxine and free T3 and TSH) measured and correlated to the changes in metabolic rate and BAT activity, given that BAT development is dependent on T3?
  3. Apart from the liver, did the authors examine muscle tissues for any increase in intramyocellular fat accumulation among the Utx KO mice?  

Author Response

 “I think a brief description of how Utx alters chromatin structure and enhances the transcription of genes relevant to adipocyte biology would be good in the Introduction.”

We added in the Introduction a description on the role of Utx in the regulation of brown adipocyte thermogenic program. “We previously reported that ubiquitously transcribed tetratricopeptide (Utx), a histone demethylase that specifically demethylates histone 3 lysine 27 di- or tri-methylation (H3K27me2 or H3K27me3) [15], promotes brown adipocyte thermogenic gene expression such as UCP1 and Pgc1alpha via a dynamic epigenetic event involving H3K27me3 demethylation followed by H3K27 acetylation [14]. Our very recent study further determined the physiological significance of Utx in vivo using a brown fat Utx knockout mouse and discovered that Utx deficiency in BAT impairs cold-induced thermogenesis, reduces energy expenditure and exacerbates diet-induced obesity”.

“For the assessment of BAT activity, could they have employed PET-CT scanning or infrared thermography or even fat fraction MRI to determine the activation of BAT in vivo in real time during cold exposure?”

Thanks for the suggestion. It would be extremely informative to measure brown fat temperature in real time during cold exposure. We unfortunately don’t have devices to conduct experiments as such in the lab.

Was the thyroid function (ie. plasma free thyroxine and free T3 and TSH) measured and correlated to the changes in metabolic rate and BAT activity, given that BAT development is dependent on T3?

Thanks for the suggestion. To maintain a well functional brown fat, the body requires two key factors. One is external drivers outside brown fat such as the sympathetic innervation and thyroid hormones. The other is internal machinery such as genetic/epigenetic pathways that autonomously regulate brown fat development and function. We agree with the Reviewer that the thyroid hormone T3 is one of the most important external drivers (besides sympathetic innervation) in the regulation of brown fat thermogenesis. In our present study, however we focus on adipocyte Utx, an internal epigenetic factor that is out of the realm of the external drivers, in the regulation of brown/white fat metabolism. As such we think that adipocyte Utx may not directly alter circulating T3 unless compensatory/secondary/feedback regulation takes place.

“Apart from the liver, did the authors examine muscle tissues for any increase in intramyocellular fat accumulation among the Utx KO mice?” 

Thanks again for the great suggestion. We don’t have skeletal muscle samples properly stored for staining intramyocellular lipid. We suspect no functional change on skeletal muscle as we did not observe any change in locomotive activity of AUTXKO mice.

Reviewer 3 Report

In this manuscript, the authors investigated the role of  histone demethylase Utx in the regulation of white adipose tissue metabolism, using  mice with Utx knockout in mature adipocytes (AUTXKO). These mice  fed a standard diet showed higher body weight, fat mass and impaired glucose tolerance. On the contrary, feeding with a high-fat diet led to a decrease in adipose tissue lipids, increased inflammation and fibrosis,  liver steatosis and impaired glucose tolerance. The authors conclude that Utx deficiency in mature while adipocytes  plays an important role in the regulation  of lipid metabolism during the development of obesity.

The topic of the study is relevant   because the epigenetic regulation of energy metabolism in obesity is given attention in connection with the increasing prevalence of obesity. The results are presented in a clear style and in a  logical way.

The reviewer has following comments to manuscript:

In method: No information is given  on how blood samples were taken to determine glucose and insulin concentrations. This is important because, for example, hemolytic serum distorts insulin levels when taken from the tail vein. The large dispersion of concentrations is due to the hemolytic plasma that occurs when taken blood from the tail vein. It is known. that hemolytic plasma/serum increases insulin concentrations as measured by ELISA.

The main shortcoming of the study is that the effect of Utx deficiency on the effects of standard and high-fat diets is observed in age-different groups (standard diet age 30 weeks,; HFD age 15 weeks). Although each group had its own control group, this difference needs to be explained. Is it possible that an over-fat diet was toxic to Utx adipocyte-deficient animals due to a lipid storage disorder in adipose tissue? In addition, in the methods section (line 77) it is stated that the diets were administered for 24 weeks. The data should be specified.

The authors state in the description under Fig. 1 and Fig.2 state that: "5-week old male AUTXKO and their littermate control fl / Y mice were put on a regular chow diet for 36 weeks." What diet did the animals have from weaning (2 weeks) to 5 weeks? Probably also chow diet, please explain.

It is common to report the weight of the fat pads and the organ as the liver not in absolute numbers but as relative to the animal's body weight. This facilitates group comparisons and the detection of obesity or hepatomegaly. The authors should report the weight of fat depots and liver in Figures 1 and 5 as an index of body weight. This method of calculation can significantly affect the data on the influence of nutritional factors and Utx deficiency on the weight of adipose tissue depots  and liver in the tested groups.

Figures 1 and 5: Also, food intake expressed in absolute grams is of no value. The authors must express relative food intake, calculated per 100 g of body weight.

Eliminate the discrepancy between the writing Fig. 2B on line 154 and the description of Fig.2 on line 165 where only Fig.2 is shown.

Fig. 3 and in Figs. 6 must be shown on the graphs showing the AUC for glucose concentrations during GTT and ITT units on the "Y" axis - mmol / L / 2 h.

Discussion

Many details, such as the activity of transport proteins involved in lipid transport across membranes, levels of circulating free fatty acids, lipoprotein lipase activity  are missing for the correct interpretation of the obtained data. or data on fatty acid release from adipose tissue.

The authors state that:“ Exaggerated adipose inflammation coupled with hepatic steatosis and NASH would lead to obesity-induced metabolic dysfunction such as insulin resistance and represent a metabolic unhealthy state, as we observed in AUTXKO mice“. Hepatic steatosis leads to NASH in only about 10%. and oxidative stress is thought to be a key factor for its development.

In the discussion, I recommend removing the notion that Utx-deficiency in adipocytes may represent metabolically unhealthy obesity. The authors do not have enough data for this claim and, moreover, the question of the existence of healthy obesity has not been confirmed.

In conclusion, the manuscript is very descriptive and data on lipid transport and circulating lipid levels are missing to assess the importance of the Utx deficiency in adipocytes in the pathophysiology of obesity and  related metabolic disorders.

Author Response

“In method: No information is given on how blood samples were taken to determine glucose and insulin concentrations. This is important because, for example, hemolytic serum distorts insulin levels when taken from the tail vein. The large dispersion of concentrations is due to the hemolytic plasma that occurs when taken blood from the tail vein. It is known. that hemolytic plasma/serum increases insulin concentrations as measured by ELISA.”

Thanks for the technical insight. We did not measure insulin in the present study. We did measure circulating triglyceride levels but we drew the blood from the terminal cardiac puncture. We measured blood glucose via tail nick during GTTs and ITTs but that is a common practice in the field when people conduct GTTs and ITTs.

“The main shortcoming of the study is that the effect of Utx deficiency on the effects of standard and high-fat diets is observed in age-different groups (standard diet age 30 weeks, HFD age 15 weeks). Although each group had its own control group, this difference needs to be explained. Is it possible that an over-fat diet was toxic to Utx adipocyte-deficient animals due to a lipid storage disorder in adipose tissue? In addition, in the methods section (line 77) it is stated that the diets were administered for 24 weeks. The data should be specified.”

Thanks for the comment. The reason that the cohorts of mice at different ages were used in chow and high-fat diet (HFD) feeding studies is explained below.

  1. The obese phenotype we observed in chow-fed AUTXKO mice was age-dependent. The knockout mice did not become obese until they approached 30-week old. We presented a figure of time dependent body weight growth in Figure 1A (left panel). The data indicate that aging process promotes obesity in AUTXKO mice on chow diet.
  2. HFD feeding however accelerated the process of weight gain in AUTXKO mice. The body weight started diverging after a two-week of HFD feeding, with AUTXKO mice staying on a higher weight throughout the duration of the experiment. We terminated the HFD feeding study after 10-week of HFD feeding because we observed a stable and constant phenotype in AUTXKO mice along the course. In a sense, while both aging and HFD promote obesity in AUTXKO mice, HFD feeding markedly shorten the obesity-developing process that takes longer time for aging to attain the outcome.
  3. Utx deletion in adipocytes limited the ability of adipocytes to store lipid evident by a decreased lipogenic program, which was further exacerbated by increased inflammation and fibrosis, a hallmark of adipose tissue remodeling and dysfunction. Nutrient (fat and glucose) influx by HFD feeding further accelerated the process of remodeling, resulting in spillover of lipids to liver from adipocytes that are marred by metabolic remodeling and dysfunction.
  4. “24 weeks” HFD feeding was an error. Now it is corrected as “10 weeks” in the Materials and Methods.

“The authors state in the description under Fig. 1 and Fig.2 state that: "5-week old male AUTXKO and their littermate control fl / Y mice were put on a regular chow diet for 36 weeks." What diet did the animals have from weaning (2 weeks) to 5 weeks? Probably also chow diet, please explain.”

Thanks for your comment. We actually weaned the mice at the age of 3 weeks directly on the chow diet and maintained the animals on the same diet throughout the experiment. We corrected the description in the legends accordingly.

“It is common to report the weight of the fat pads and the organ as the liver not in absolute numbers but as relative to the animal's body weight. This facilitates group comparisons and the detection of obesity or hepatomegaly. The authors should report the weight of fat depots and liver in Figures 1 and 5 as an index of body weight. This method of calculation can significantly affect the data on the influence of nutritional factors and Utx deficiency on the weight of adipose tissue depots and liver in the tested groups.”

Thanks for the suggestion. We now normalized the organ weight to the respective body weight and found that the results on the difference of the organ weight between the two genotype groups remained the same. We now presented the data of normalized organ weight along with the original figures (with the absolute weight values) in Figure 1C and 5C.

“Figures 1 and 5: Also, food intake expressed in absolute grams is of no value. The authors must express relative food intake, calculated per 100 g of body weight.”

We now normalized the food intake to 100g of the respective body weight as suggested and presented the data of relative food intake along with the original figures (with the absolute amount of daily food intake per mouse) in Figure 1G and 5G

“Eliminate the discrepancy between the writing Fig. 2B on line 154 and the description of Fig.2 on line 165 where only Fig.2 is shown.”

Corrected. Fig. 2 was now stated on page 3, line 146.

Fig. 3 and in Figs. 6 must be shown on the graphs showing the AUC for glucose concentrations during GTT and ITT units on the "Y" axis - mmol / L / 2 h.

Now the units on the Y axis of Fig. 3 and Fig. 6 were corrected as “mg/dl/2h”.

Discussion

“Many details, such as the activity of transport proteins involved in lipid transport across membranes, levels of circulating free fatty acids, lipoprotein lipase activity are missing for the correct interpretation of the obtained data. or data on fatty acid release from adipose tissue.”

Thanks for the comments. These are all great points that can potentially explain the phenotype where lipid storage is shifted from adipose tissue to liver. Although we don’t have resources to conduct all the experiments as suggested, we did measure phosphor-HSL in iWAT of AUTXKO and their littermate controls. There was a tendency towards increase of phosphor-HSL, albeit not reaching statistical significance, in iWAT of AUTXKO mice (the results were presented in Supplemental Figure 2 and 4), suggesting a possibility of increased release of free fatty acids from adipocytes. We also have data indicating increased circulating triglyceride levels in AUTXKO mice (Fig. 8B). Nonetheless, future studies involving a careful examination of circulating fatty acids, fatty acid transport in the liver and lipoprotein lipase activity will be warranted to depict the exact pathways underlying the shift of lipid storage from adipose tissue to liver in AUTXKO mice. This has been added to the discussion on page 10, line 327-340.

“The authors state that:“ Exaggerated adipose inflammation coupled with hepatic steatosis and NASH would lead to obesity-induced metabolic dysfunction such as insulin resistance and represent a metabolic unhealthy state, as we observed in AUTXKO mice“. Hepatic steatosis leads to NASH in only about 10%. and oxidative stress is thought to be a key factor for its development.”

Thanks for the insightful comment on NASH. By removing “NASH” and “represent a metabolic unhealthy state”, now we stated “Exaggerated adipose inflammation coupled with hepatic steatosis would lead to obesity-induced metabolic dysfunction such as insulin resistance”.

“In the discussion, I recommend removing the notion that Utx-deficiency in adipocytes may represent metabolically unhealthy obesity. The authors do not have enough data for this claim and, moreover, the question of the existence of healthy obesity has not been confirmed.”

Thanks for the comment. We deleted the statement on “metabolically unhealthy obesity” as suggested.

“In conclusion, the manuscript is very descriptive and data on lipid transport and circulating lipid levels are missing to assess the importance of the Utx deficiency in adipocytes in the pathophysiology of obesity and related metabolic disorders.”

As we stated above, we agree with the Reviewer that future studies involving a careful examination of circulating fatty acids, fatty acid transport in the liver and lipoprotein lipase activity will be warranted to depict the exact pathways underlying the shift of lipid storage from adipose tissue to liver in AUTXKO mice. The discussion on future studies has been added to the section of Discussion on page 10, line 327-340.

Round 2

Reviewer 1 Report

My comments have been addressed and I don't have further comments.